

# Ferrostatin-1 post-treatment attenuates acute kidney injury in mice by inhibiting ferritin production and regulating iron uptake-related proteins

Yanxiu Zhao[1], Binhua Jiang[2], Dinghui Huang[3], Juxiang Lou[1], Guoshun Li[1], Jianqi Liu[1], Fuhui Duan[1], Yuan Yuan[4] and Xiaoyan Su[1]

[1] Department of Nephrology, Baoshan People's Hospital, Baoshan, People's Republic of China
[2] Department of Obstetrics, Baoshan People's Hospital, Baoshan, People's Republic of China
[3] Department of Pediatrics, Baoshan People's Hospital, Baoshan, People's Republic of China
[4] Intensive Care Unit, Ningbo Medical Center Lihuili Hospital, Ningbo, People's Republic of China

Corresponding authors
Yuan Yuan,
yuanyuan1040@163.com
Xiaoyan Su,
Su13987058543@163.com

## ABSTRACT

**Background:** Acute kidney injury (AKI) is a common and serious medical condition with high morbidity and mortality. Recent research has highlighted ferroptosis, a novel form of programmed cell death, as a potential therapeutic target in mitigating renal tubular injury in AKI. Ferrostatin-1, a specific ferroptosis inhibitor, has been demonstrated to prevent renal injury through ferroptosis inhibition.

**Methods:** Utilizing a murine AKI model, we investigated the effects of Ferrostatin-1 by administering it post-injury. Through high-throughput sequencing and pathological analysis, we focused on the critical role of ferroptosis-related pathways in the treatment.

**Results:** Ferrostatin-1 post-conditioning effectively mitigated oxidative damage and reduced iron content associated with AKI. Additionally, critical ferroptosis-related proteins, such as GPX4, SLC7A11, NRF2, and FTH1, exhibited increased expression levels. *In vitro*, Ferrostatin-1 treatment of HK-2 cells significantly diminished lipid peroxidation and iron accumulation. Furthermore, Ferrostatin-1 was found to downregulate the PI3K signalling pathway.

**Conclusion:** Ferrostatin-1 acted as a potential ferroptosis inhibitor with the capacity to enhance antioxidant defences. This study suggests that Ferrostatin-1 could serve as a promising novel strategy for improving the treatment of AKI and promoting recovery from the condition.

## INTRODUCTION

Acute kidney injury (AKI), characterized by a rapid deterioration in renal function, is a widespread and serious health issue accompanied by high morbidity and mortality rates. Frequently secondary to extrarenal events, AKI impacts nearly 30% of critically ill patients and 5% of hospitalized patients (*Bellomo, Kellum & Ronco, 2012*). However, the molecular mechanisms underpinning AKI are not fully understood, and existing preventive and therapeutic strategies remain inadequate. Hence, the identification of potent

pharmaceutical agents capable of preserving kidney function and improving the prognosis of AKI patients in clinical scenarios is of paramount importance.

Ferroptosis, a recently identified form of cellular death, differs from necrosis, exhibiting a distinct morphological phenotype. This pathway is marked by an accumulation of lipid peroxides and heightened levels of free iron (*Dixon et al., 2012*). Implicated in a multitude of diseases, including Parkinson's, Alzheimer's, cancer, stroke, ischemia-reperfusion injury, intracranial hemorrhage, and brain injury, ferroptosis has garnered significant attention. An emerging body of research has suggested a potential link between ferroptosis and AKI. Some studies have demonstrated that renal tubular cells isolated *in vitro* exhibit heightened sensitivity to ferroptosis, suggesting ferroptosis could serve as an effective therapeutic target for diseases associated with renal tubular cell death (*Linkermann et al., 2013*, *2014*). Furthermore, studies have elucidated the relationship between AKI and iron metabolism (*Scindia et al., 2015*), while others have proposed that iron-mediated cell death may be the principal driving factor in various ischemic injury models (*Tonnus & Linkermann, 2017*).

Glutathione peroxidase 4 (GPX4), a primary indicator of ferroptosis, plays a crucial role in decreasing lipid peroxide, necessitating glutathione (GSH) as the requisite substrate (*Friedmann Angeli et al., 2014*; *Yang et al., 2014*; *Ursini & Maiorino, 2020*). System Xc– is an amino acid reverse transport protein that exists in the cell membrane and is composed of SLC7A11 and SLC3A2 heterodimers. The Xc– transport system manages the reciprocal exchange between intracellular glutamate and extracellular cysteine (*Bridges, Natale & Patel, 2012*). Cystine could be converted into cysteine in the cell and becomes the synthesis substrate of GSH. GSH further cooperates with GPX4 to exert an antioxidant effect and maintain cellular redox balance. Therefore, the inactivation or abnormal expression of system Xc– will affect the ability of cells to process ROS. Studies have found that if the total amount of amino acids (especially cystine) in serum were deficient *in vitro*, it would cause rapid cell death. Later, it was confirmed that this type of cell death is ferroptosis (*Gao et al., 2015*). Early studies have confirmed that system Xc– is abundant in the brush border membrane of renal tubular cells (anatomically, the part where the kidney mainly carries out amino acid transport) (*Burdo, Dargusch & Schubert, 2006*). Inhibition of system Xc– or ischemic state will lead to the lack of amino acid supply, intracellular GSH level decreased, and the antioxidant capacity diminished, leading to ferroptosis and renal tubular injury. Moreover, ferritin heavy chain 1 (FTH1) regulates intracellular iron storage. NF-E2-related factor 2 (NRF2) acts as the transcription controller for GPX4, SLC7A11, FTH1, and other proteins related to iron-induced death. These elements are deemed as the biomarkers of ferroptosis.

Recent research has illustrated the critical role of ferroptosis in modulating AKI. Evidence of ferroptosis has been observed in various types of AKI, including those induced by cisplatin, rhabdomyolysis, ischemia/reperfusion, or folic acid. The aim of this study was to ascertain whether Ferrostatin-1, a potential ferroptosis inhibitor, could mitigate AKI by reversing ferroptosis and diminishing lipid peroxidation and iron accumulation in both *in vitro* and *in vivo* models. We further sought to uncover the involved pathway through

transcriptome analysis, with the hope of offering a novel strategy for the clinical management of AKI.

## MATERIALS AND METHODS

### Cell culture and treatment

The HK-2 cells, epithelial cells derived from the proximal tubule of the human kidney, were maintained in Dulbecco's Modified Eagle Medium: F-12 (DMEM/F-12) fortified with 10% Fetal Bovine Serum (FBS) (Gibco, Billings, MT, USA). These cells were cultivated in an environment composed of 95% air and 5% $CO_2$, regulated at a constant temperature of 37 °C. For each experiment, HK-2 cells were seeded into 96-well culture plates at a density of $5 \times 10^5$ cells/well and cultured for 24 h. After this incubation period, the HK-2 cells occupied approximately 80% of the plate. Subsequently, Lipopolysaccharide (LPS) (10 μg/ml) was added to the cells and incubated for 22 h to establish the LPS-induced AKI cell model (*Sun et al., 2021*). Then, Ferrostatin-1 (1 μM) was introduced to the HK-2 cells as per the protocol outlined in a prior study (*Qiang et al., 2020*). For the study's purpose, the HK-2 cells were randomly assigned to one of three groups: the control group, the AKI model group induced by LPS at a concentration of 10 μg/ml, and the group exposed to both LPS (10 μg/ml) and Ferrostatin-1 (1 μM).

### Animals and experimental protocol

The animal experiments were approved by the Ethics Committee of Baoshan People's Hospital (SYSU-IACUC-2020-A0327). For this investigation, we secured a total of 18 C57BL/6 mice, each weighing between 20–25 g. They were accommodated in a regulated habitat with a 12-h light-dark cycle and unrestricted access to standard diet and water. To guarantee uniformity, the environmental temperature was sustained at 25 °C and relative humidity was regulated between 40% and 70%.The study was approved by the Animal Ethics Committee of Baoshan People's Hospital and conducted accordingly. The mice were divided randomly into three groups ($n = 6$ per group): the Sham group, the CLP group, and the CLP + Ferrostatin-1 group. The CLP-induced AKI model *in vivo* was established as previously described (*Miyaji et al., 2003*).

To induce AKI, mice were subjected to cecal ligation and puncture (CLP) at a distance of 5 mm from the cecal tip. This was followed by gentle puncturing of the cecum twice, and squeezing out 0.5 mm of fecal material using a 22-gauge needle. Mice that underwent the identical protocol but without the cecal ligation or puncture constituted the sham group. All mice were given an intraperitoneal dose of pentobarbital at 50 mg/kg. This was followed by subcutaneous injection of normal saline at a dose of 1 ml per 25 g of body weight. After 5 h post-CLP surgery, either normal saline or 1.5 mg/kg of Ferrostatin-1 (Sigma-Aldrich, St. Louis, MO, USA) was intravenously injected. Twenty-four hours post-injection, the animals were humanely euthanized, and their kidney and blood samples were harvested for subsequent analysis.

Animal care, including feeding, housing, and enrichment, was carried out according to established guidelines. Criteria for euthanizing animals were established prior to the

planned end of the experiment. Any surviving animals at the conclusion of the experiment were monitored and accounted for accordingly.

## BUN and Cr detection

The serum gathered from the blood samples by centrifugation (3,000 rpm, 5 min, 4 °C) was used to evaluate renal function. The levels of serum BUN and Cr (Creatinine) were detected by the commercial assay kit—Urea assay kit (Nanjing Jiancheng, Nanjing, Jiangsu, China) and Creatinine assay kit (sarcosine oxidase) (Nanjing Jiancheng, Nanjing, Jiangsu, China), according to the manufacturer's instructions, respectively.

## ROS detection

The level of ROS in the renal tissues was determined by Dihydroethidium (DHE) fluorescence with a DCFH-DA assay kit. Renal tissues were stored at −20 °C until the fluorescence assay. The blocks were sliced onto glass slides coated with polylysine and then treated with 10 μM DHE at 37 °C for 30 min. The samples were then incubated with DAPI solution at room temperature for 10 min, kept away from light. Following this, the tissues underwent a series of three washes with PBS at room temperature. After the washes, the samples were incubated with DCFH-DA staining solution at 37 °C for 1 h. Subsequently, fluorescent microscopy was employed to acquire images.

The previously mentioned assay kit was used to evaluate the ROS level in HK-2 cells, adhering to the guidelines provided by the manufacturer. The cells were maintained in a controlled setting at 37 °C with an atmosphere of 95% air and 5% $CO_2$. They were placed on 35 mm laser confocal Petri dishes, with a seeding density of $1.0 \times 10^5$ cells per well, and cultured for a period of 24 h. The HK-2 cells were treated with Ferrostatin-1 (1 μM) for 24 h and then were treated with 10 μM DCFH-DA for 20 min in the dark. Following the aforementioned process, the HK-2 cells were thoroughly rinsed with PBS three times to eliminate any remaining DCFH-DA. Then the images of HK-2 cells were captured with the fluorescence microscope.

## Histopathological analysis

Paraformaldehyde was used to fix the dissected renal samples, and then paraffin was used to embed the samples. The 4-μm slices were stained with HE (*Chen et al., 2011*). The specimens were evaluated in a blinded manner.

## Quantitative real-time polymerase chain reaction

Real-time PCR (polymerase chain reaction) of HK-2 cells and kidney samples was performed (*Kang et al., 2001*). For the PCR reaction, a volume of 20 μl was used, consisting of 30 ng RNA as the template and 12 μl SYBR Green PCR Master Mix (Applied Biosystems). The following primer sequences were used: Gpx4 forward primer 5′-CCGGCTACAATGTCAGGTTT-3′ and reverse primer 5′-ACGCAGCCGTTCTTATCAAT-3′; SLC7A11 forward primer 5′-GATGCTGTGCTTGGTCTTGA-3′ and reverse primer 5′-GCCTAC CATGAGCAGCTTTC-3′; NRF2 forward primer 5′-CTTTTATCTCACTTTACCGCCCGAG-3′ and reverse primer 5′-GACACGTGGGAGTTCAGAGGG-3′; and FTH1 forward primer 5′-

ATGATGTGGCCCTGAAGAAC-3′ and reverse primer 5′-TCATCACGGTCAGGTTTCTG-3′. The PCR was carried out using the following conditions: 5 min at 95 °C, followed by 36 cycles of 30 s at 94.5 °C, 30 s at 60 °C, and 60 s at 72 °C. The final extension was performed at 72 °C for 7 min. The comparative CT method with $2^{-\Delta\Delta Ct}$ was used to calculate the relative expression of the genes.

## Western blot analysis

Western blot analysis was employed to examine the proteins extracted from renal tissues or HK-2 cells, following previously described methods (*Fei et al., 2020*; *Zhang et al., 2016*). Initially, proteins were lysed with radioimmunoprecipitation assay (RIPA) buffer, collected, and their concentration determined. Subsequently, the samples were subjected to electrophoresis in a 10% Sodium dodecyl sulfate (SDS)-polyacrylamide gel. Post-electrophoresis, proteins were transferred onto a polyvinylidene fluoride (PVDF) membrane. This membrane was then blocked with Tris-buffered saline with Tween 20 (TBST) solution containing 5% nonfat milk powder for a period of 2 h. The membrane was then incubated with the primary antibody at a temperature of 4 °C overnight. This was followed by an incubation with the secondary antibody at room temperature for another 2 h. The membrane was subsequently washed and a chemiluminescent reagent was applied. The resultant band images were captured using ImageJ gel analysis software. Finally, the relative densities of the detected protein bands were quantified.

## Determination of GSH and MDA content

MDA (malondialdehyde) and GSH (glutathione) levels were determined using a commercially available assay kit from Nanjing Jiancheng, China, in accordance with the manufacturer's instructions. The levels of MDA and GSH were detected using a microplate fluorometer at 532 and 405 nm, respectively, as directed by the manufacturer's instructions.

## Iron assay

A commercial iron assay kit (Sigma-Aldrich, St. Louis, MO, USA) was used to detect the total amount of iron. To prepare the renal tissue samples for analysis, they were homogenized in iron assay buffer, following the manufacturer's instructions. Then centrifuged at 15,000 g for 8 min to collect supernatant. A total of 10 µl supernatant and 90 µl iron analysis buffer and 5 µl iron reductant were mixed and incubated for 30 min. Then, the total iron levels were measured with an iron probe at 593 nm.

HK-2 cells were also homogenized in iron assay buffer and centrifuged along with standards. A total of 10 µl supernatant, 90 µl iron analysis buffer and 5 µl iron reductant were mixed and incubated. After 30 min, the total iron levels were measured at 593 nm with the iron probe.

## Transcriptome sequencing

Samples obtained from the Ferrostatin-1 (Fer-1) and Lipopolysaccharide (LPS) groups ($n = 3$) were flash-frozen in liquid nitrogen. A third-party company executed total RNA extraction and the subsequent sequencing process. High-throughput sequencing was

employed to analyze differentially expressed genes (DEGs) and identify associated pathways.

## Statistical analysis

Data are represented as mean ± standard deviation (SD) and were analyzed using the Statistical Package for the Social Sciences (SPSS) software (SPSS Inc. Chicago, IL, USA). For comparing the means between two groups, the unpaired two-tailed Student's t-test was employed. Additionally, differences among multiple groups were assessed using one-way analysis of variance (ANOVA) and Dunnett's multiple comparison tests. A $P$ value of < 0.05 was considered to indicate statistical significance. The figures were generated using GraphPad Prism Software. All experiments were performed with a minimum of three biological replicates.

## RESULTS

### Ferrostatin-1 post-treatment protects against AKI in the mice model

The impact of Ferrostatin-1 on the AKI was confirmed by adding Ferrostatin-1 to the CLP group. Figure 1A shows significant pathological changes in the CLP model. The renal tissue morphology appeared to be normal in the Sham group, displaying a clear structure with regularly arranged cells, and no signs of inflammation or fiber exudation were observed; while in the CLP group, there was brush-like edge damage, proximal tubular dilation, interstitial widening, proteinaceous casts and necrosis. Compared with the CLP group, the Fer-1group showed alleviated pathological changes of brush-like edge damage, proximal tubular dilation, interstitial widening, proteinaceous casts and necrosis. According to the data presented in Figs. 1C and 1D, the concentration of serum BUN and Cr in the CLP group was found to be significantly higher than that in the Sham group ($P < 0.05$). However, after administering Ferrostatin-1, the levels of serum BUN and Cr were observed to decrease, although they still remained higher than those in the Sham group ($P < 0.05$). These results suggested that Ferrostatin-1 could significantly mitigate kidney injury.

### Ferrostatin-1 post-treatment attenuates ferroptosis during AKI

MDA, GSH and ROS were measured to prove that Ferrostatin-1 had protection on AKI induced by Ferroptosis. In the CLP group, ROS level and MDAlevel increased while GSH level decreased. After treatment with Ferrostatin-1, the oxidative damage caused by AKI was inhibited, as evidenced by a decrease in levels of ROS and MDA, and an increase in the level of GSH, when compared to the CLP group ($P < 0.05$) (Figs. 1B, 1E and 1F). Secondly, the iron content and the expressions of GPX4, SLC7A11, NRF2 and FTH1 were valued. In the AKI model group, the iron content (Fig. 1G) increased ($P < 0.05$) while the expressions of GPX4, SLC7A11, NRF2 and FTH1 decreased ($P < 0.05$) (Fig. 2). However, post-treatment with Ferrostatin-1 suppressed these alterations ($P < 0.05$) (Fig. 2). These finding suggested that post-treatment with Ferrostatin-1 mitigated AKI by inhibiting ferroptosis.

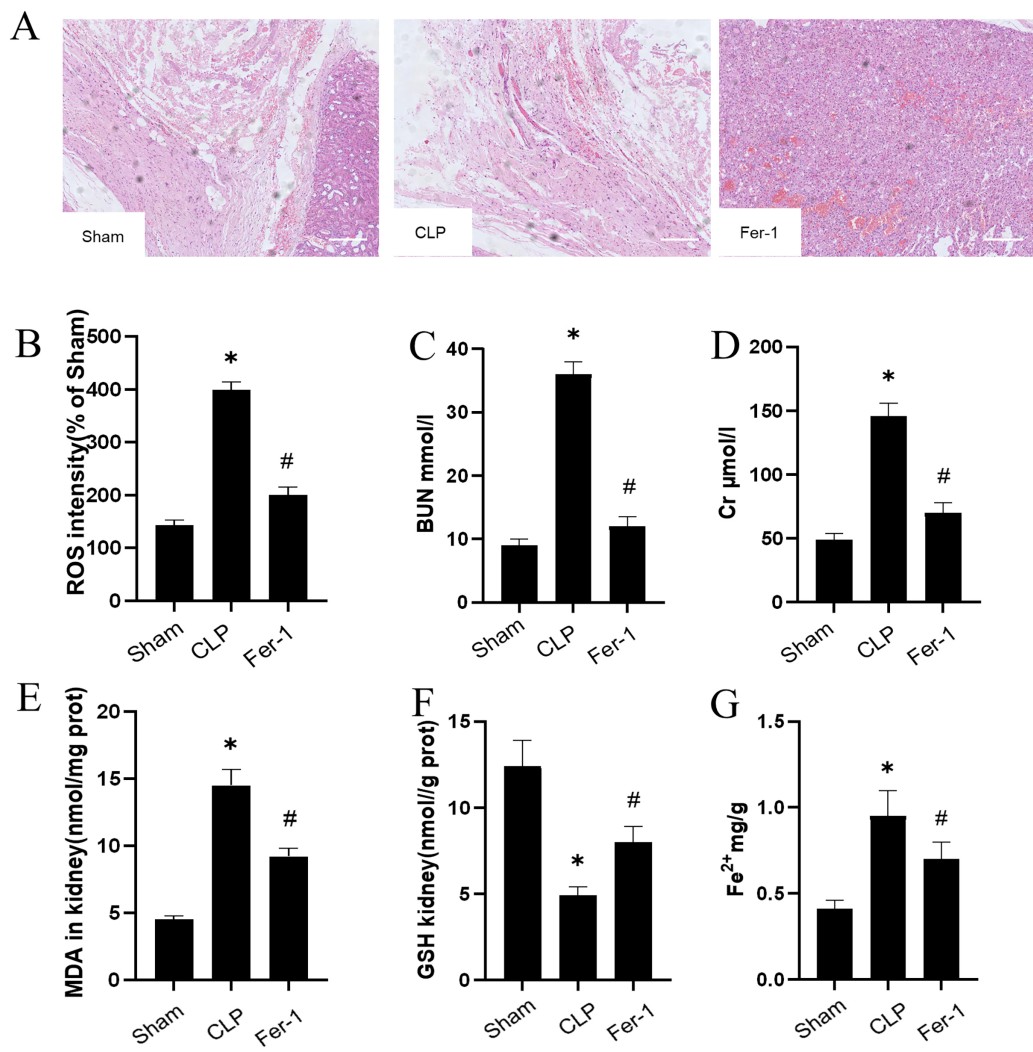

**Figure 1 Ferrostatin-1 attenuates CLP-induced kidney injury.** Mice were given a sham operation or CLP model operation, with or without ferrostatin-1 (Ferrostatin-1) administration. (A) Representative images (200×) showing HE staining of renal sections (scale bar = 50 μm). (B) The quantification of the DHE fluorescence intensity of ROS. (C) BUN concentration in the plasma. (D) Cr concentration in the plasma. The levels of MDA (E), GSH (F), and $Fe^{2+}$ (G) in mouse kidney homogenates. $^{*}P < 0.05$ *vs.* Sham group, $^{\#}P < 0.05$ *vs.* CLP group. Data are presented as mean ± SD.

## Ferrostatin-1 mitigates ferroptosis in LPS-induced HK2 cells

Drawing upon the aforementioned *in vivo* results, we proceeded to assess the impact of Ferrostatin-1 on LPS-induced AKI. As illustrated in Fig. 3A, significant pathological alterations were observed in the LPS-induced AKI model, including brush border disruption, dilation of proximal tubules, widening of interstitial spaces, formation of proteinaceous casts, and necrosis. As shown in Fig. 3B, post-treatment with Ferrostatin-1 induced ROS levels in the LPS-induced AKI group (*P* < 0.05). Similarly, post-treatment

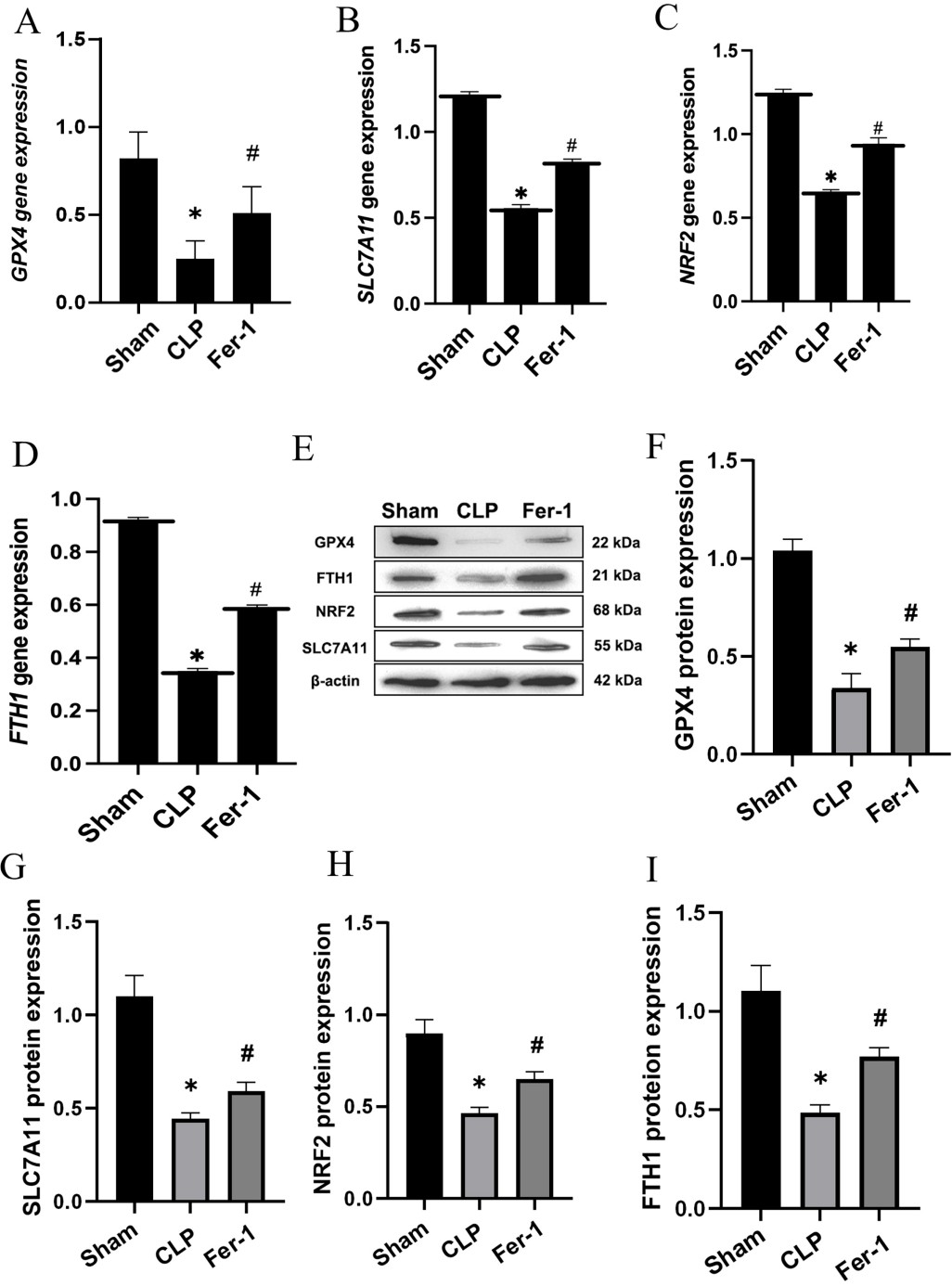

**Figure 2  Ferrostatin-1 alleviates the occurrence of ferroptosis in AKI.** (A–D) Quantitation of *GPX4*, *SLC7A11*, *NRF2* and *FTH1* gene expression. (E) Western blot analysis of GPX4, SLC7A11, NRF2 and FTH1 proteins in the kidney tissue ($n = 3$). (F–I) Quantitation of GPX4, SLC7A11, NRF2 and FTH1 protein expression. $^*P < 0.05$ *vs*. Sham group, $^#P < 0.05$ *vs*. CLP group. Data are presented as mean ± SD.           

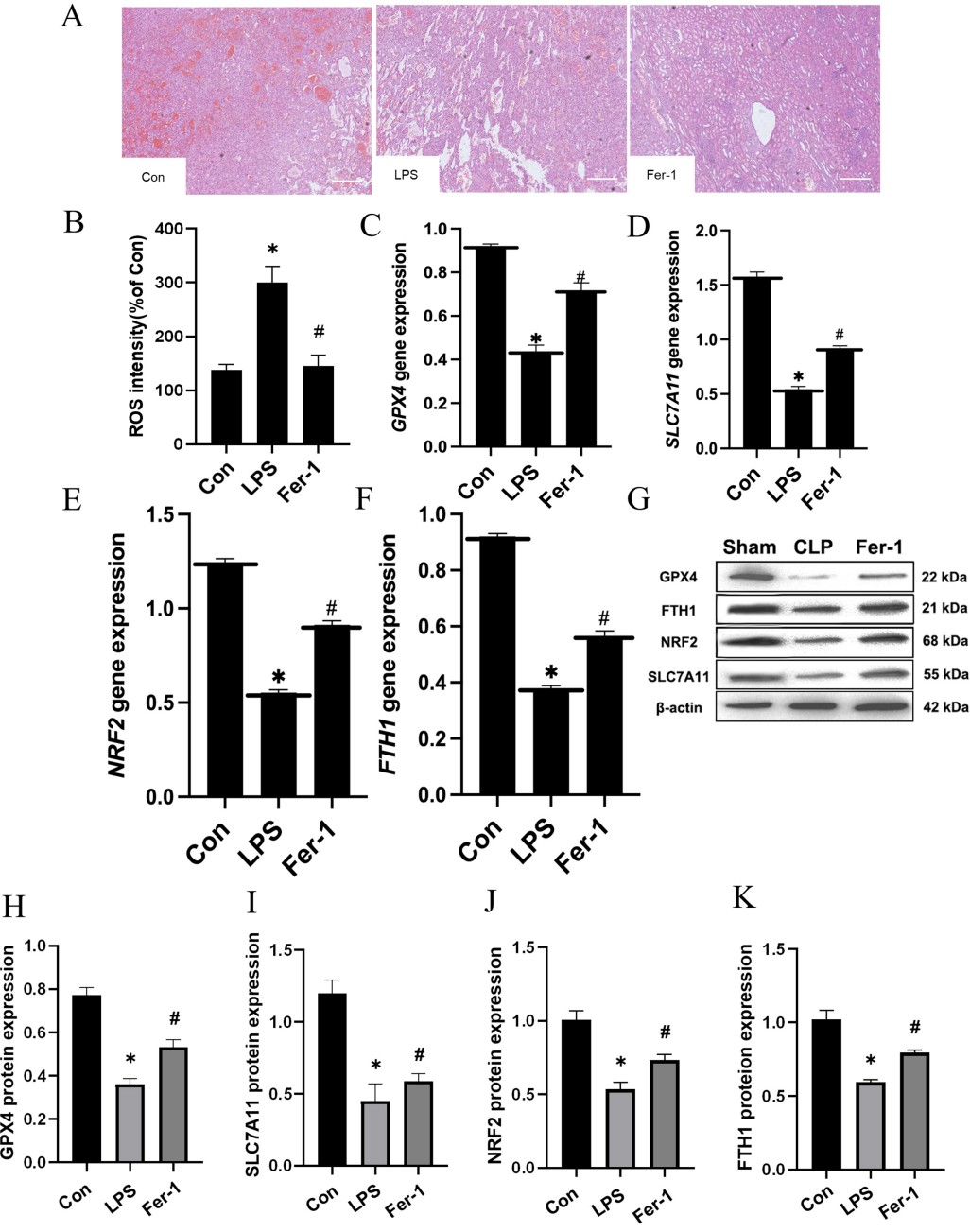

**Figure 3 Ferrostatin-1 inhibits LPS-induced ferroptosis in HK-2 cells.** (A) Representative images (200×) showing HE staining of renal sections (scale bar = 50 μm). (B) The quantification of the DCFH-DA fluorescence intensity of ROS. (C–F) Quantitation of *GPX4*, *SLC7A11*, *NRF2* and *FTH1* gene expression. (G) Western blot analysis of GPX4, SLC7A11, NRF2 and FTH1 proteins in the HK-2 cells ($n = 3$). (H–K) Quantitation of GPX4, SLC7A11, NRF2 and FTH1 protein expression. $^*P < 0.05$ *vs.* Control (Con) group, $^\#P < 0.05$ *vs.* LPS group. Data are presented as mean ± SD.

with Ferrostatin-1 prevented the decrease of the GPX4, SLC7A11, NRF2 and FTH1 expression in the Ferrostatin-1 treatment group ($P < 0.05$) (Figs. 3C–3K). Obviously, Ferrostatin-1 inhibited LPS-induced ferroptosis *in vitro*.

### Ferrostatin-1 inhibits LPS-induced ferroptosis by inhibiting ferritin production and regulating iron uptake-related proteins *in vitro*

In our quest for a holistic comprehension of the molecular mechanisms driving AKI, we implemented high-throughput transcriptional sequencing. This was to identify the differentially expressed genes (DEGs) contrasting the Ferrostatin-1 and LPS groups ($n = 3$ per group). A total of 63,677 genes were identified in the RNA-seq data set. The heatmap showed 5,099 DEGs in the Ferrostatin-1 and LPS groups (Fig. 4A). The Ferrostatin-1 group displayed 26 downregulated and 35 upregulated DEGs according to the volcano plot presented in Fig. 4B. Subsequently, we conducted GO/KEGG analysis on these DEGs and discovered that the phosphatidylinositol 3-kinase signalling (PI3K) pathway was implicated, as shown in Fig. 4C. This observation suggests a potential association between ferroptosis and AKI, and Ferrostatin-1 potentially inducing AKI in mice *via* the PI3K pathway. Consequently, these findings provide further evidence supporting the involvement of Ferrostatin-1 in ferroptosis-mediated AKI.

## DISCUSSION

The kidney, a vulnerable organ, significantly impacts septic patients' outcomes, as renal injury contributes to increased mortality rates (*Peerapornratana et al., 2019*; *Poston & Koyner, 2019*). Plenty of studies have established a correlation between lipid peroxidation and ROS in AKI. Ferroptosis, a novel iron-dependent, lipid peroxidation-induced cell death pathway, has been recently identified (*Miyaji et al., 2003*). A recent study investigated the effect of ferrostatin-1 and its potential efficacy in AKI therapy. In this study, ferrostatin-1 was used to treat CLP-induced AKI mice. The improvement of renal pathology in mice after ferrostatin-1 treatment was observed, and the changes in indicators such as ROS, MDA, GSH, BUN, and Cr were compared. The pivotal findings of this study encompass ferroptosis' involvement in AKI, the protective effect of Ferrostatin-1 post-treatment against AKI *via* ferroptosis inhibition, and Ferrostatin-1's protective mechanism's reliance on the PI3K pathway (*Peerapornratana et al., 2019*; *Poston & Koyner, 2019*; *Fani et al., 2018*).

Recent investigations have revealed that the inhibition of GPX4 could diminish the reduction of lipid peroxides (*Lei, Bai & Sun, 2019*). GPX4 utilizes GSH and eliminates lipid peroxides (*Hadian & Stockwell, 2020*). Several studies have demonstrated that Ferrostatin-1 can inhibit ferroptosis and alleviate acute kidney injury (*Scindia Ph, Leeds Md & Swaminathan Md, 2019*; *van Swelm, Wetzels & Swinkels, 2020*). In line with these findings, our study also revealed that post-treatment with Ferrostatin-1 could mitigate AKI. Since ferroptosis occurs during AKI, Ferrostatin-1 may offer a promising strategy for AKI prevention. Although ferroptosis has been proven to have a renal protective effect in AKI, there are still some urgent issues to be addressed. Such as the mechanism by which ferroptosis is ultimately triggered remains unclear, the degree of ferroptosis involvement in the coexistence of multiple cell death modes in AKI, and whether different factors have an impact on it. These issues require further research to provide insights for the understanding and treatment of AKI and related diseases.

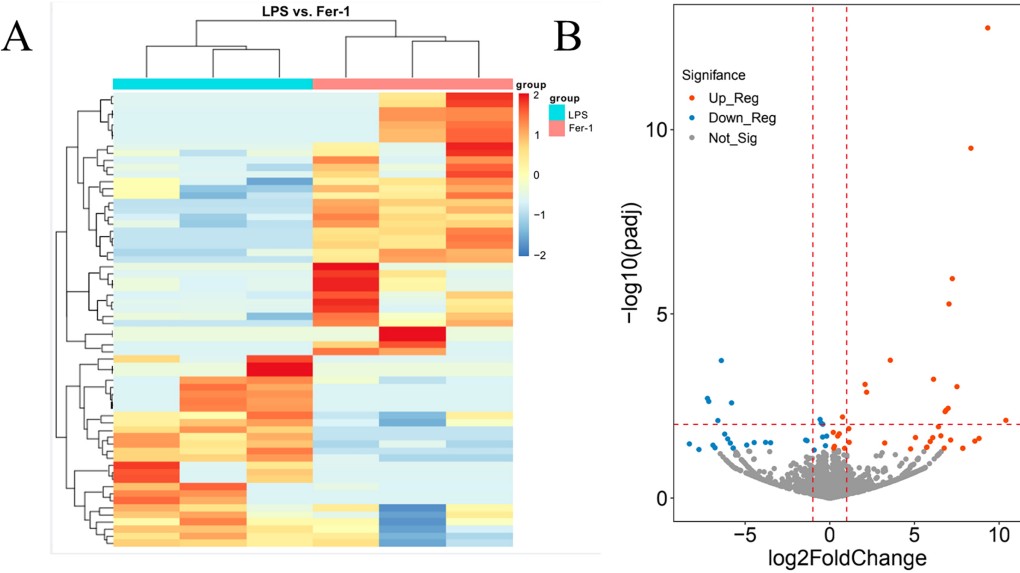

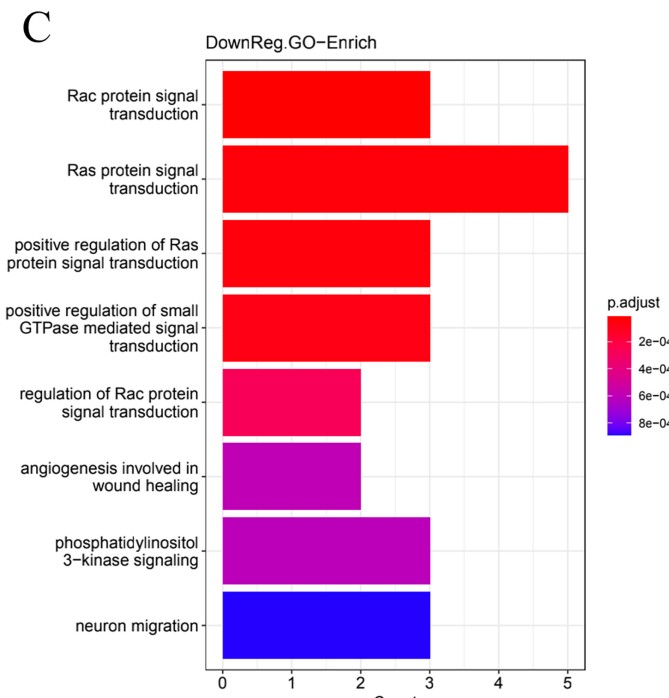

**Figure 4 HK-2 cells were involved in transcriptome sequencing.** (A) Heatmap showed the hierarchical cluster analysis of the differentially expressed genes (DEGs) in HK-2 cells. (B) Volcano plot represented the overall distribution of gene expression levels and differential multiples between the LPS group and Ferrostatin-1 group. (C) Enrichment analysis of the Gene Ontology (GO) signalling pathway in the Ferrostatin-1 group compared with the LPS group assayed by RNA-sequencing of HK-2 cells.

Moreover, the safety of Ferrostatin-1 is not completely understood since it is still under investigation in preclinical studies. Some studies have reported potential side effects of Ferrostatin-1, such as potential liver toxicity, weak cytoprotective effects, and induction of

autophagy. Ferrostatin-1 may cause liver damage in high doses. It is believed that oxidative stress and lipid peroxidation inhibition caused by Ferrostatin-1 may increase liver injury (*Gaschler et al., 2018*). Some studies have demonstrated that Ferrostatin-1 may have limited cytoprotective effects, which may be due to its inability to completely prevent lipid peroxidation (*Skouta et al., 2014*; *Badgley et al., 2020*). According to several studies (*Friedmann Angeli et al., 2014*; *Tang et al., 2020*) Ferrostatin-1 has been found to potentially induce autophagy, a complex cellular process that plays a role in maintaining cellular homeostasis and can have varying effects depending on the specific context. Ferrostatin-1 inhibits lipid peroxidation by interacting with and sequestering lipid hydroperoxides; however, it may also affect other cellular pathways and result in off-target effects (*Gaschler et al., 2018*; *Tang et al., 2020*). Other researchers have reported potential effects on cell signaling pathways and immune function, which could have implications for safety and efficacy (*Yoshida, 2017*; *Bonventre & Yang, 2011*). Given these findings, it is clear that while Ferrostatin-1 has shown significant potential in preclinical studies as a therapeutic agent and an inhibitor of ferroptosis, more research is needed to fully understand the potential risks and benefits of ferrostatin-1 before it can be considered for clinical use. This will likely involve further preclinical studies to investigate potential side effects in more detail, as well as clinical trials in humans to establish its safety and efficacy in a real-world setting.

Our study, however, is not without limitations. Primarily, our research focused on the impact of Ferrostatin-1 on ferroptosis. While we effectively demonstrated the direct anti-ferroptosis activity of Ferrostatin-1, we did not probe its potential influence on other types of cell death such as autophagy and necroptosis. Additionally, the specific mechanism through which Ferrostatin-1 modulates the PI3K pathway requires further investigation. The murine model of AKI is frequently used to study acute kidney injury in humans. It involves the induction of kidney injury in mice using nephrotoxic agents, ischemia-reperfusion injury, cecal ligation and puncture (CLP), or genetic modifications. This model offers a certain degree of reproducibility, thus enabling consistent results across multiple experiments under identical conditions. Despite its utility in studying kidney injury, the murine AKI model presents several limitations and challenges, including variability in the animal's response to the injury and potential discrepancies between murine and human kidney physiology (*Gao et al., 2016*). Furthermore, despite the model's valuable contributions, it may not entirely encompass the complexity and heterogeneity of human AKI. Consequently, it's essential to bolster animal studies with clinical research and validation. In conclusion, while the AKI mouse model offers valuable insights for kidney disease research, its limitations in reproducibility and relevance need to be acknowledged. Comprehensive analyses and interpretations should be conducted in conjunction with clinical data from humans.

## CONCLUSIONS

In summary, AKI triggers ferroptosis by increasing iron contents and lipid peroxidation. Ferrostatin-1, acting as a ferroptosis inhibitor, enhances antioxidant capacity and effectively alleviates AKI by inducing the gene expression of lipid metabolism enzymes

within the PI3K pathway. This study offers compelling evidence supporting the potential application of ferrostatin-1 in the treatment of AKI.

### Funding
The authors received no funding for this work.

### Competing Interests
The authors declare that they have no competing interests.

### Author Contributions
- Yanxiu Zhao performed the experiments, prepared figures and/or tables, authored or reviewed drafts of the article, and approved the final draft.
- Binhua Jiang analyzed the data, prepared figures and/or tables, and approved the final draft.
- Dinghui Huang analyzed the data, prepared figures and/or tables, and approved the final draft.
- Juxiang Lou analyzed the data, prepared figures and/or tables, and approved the final draft.
- Guoshun Li conceived and designed the experiments, analyzed the data, prepared figures and/or tables, and approved the final draft.
- Jianqi Liu conceived and designed the experiments, analyzed the data, prepared figures and/or tables, and approved the final draft.
- Fuhui Duan conceived and designed the experiments, analyzed the data, prepared figures and/or tables, and approved the final draft.
- Yuan Yuan conceived and designed the experiments, analyzed the data, prepared figures and/or tables, authored or reviewed drafts of the article, and approved the final draft.
- Xiaoyan Su conceived and designed the experiments, authored or reviewed drafts of the article, and approved the final draft.

### Animal Ethics
The following information was supplied relating to ethical approvals (*i.e.*, approving body and any reference numbers):

The ethical considerations of this study involving human participants were reed and approved by the Animal Ethics Committee of Baoshan People's Hospital (SYSU-IACUC-2020-A0327).

### Data Availability
The raw data is available in the Supplemental Files.

## Supplemental Information

Supplemental information for this article can be found online at http://dx.doi.org/10.7717/peerj.15786#supplemental-information.

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
