# Peer review of "Ferrostatin-1 post-treatment attenuates acute kidney injury in mice by inhibiting ferritin production and regulating iron uptake-related proteins"

_PeerJ, doi:10.7717/peerj.15786_

## Round 0.1 · original submission · Minor Revisions

A detailed point to point response letter (editor + reviewers), manuscript with tracked font are necessary for further process when they submit their revised version of paper. Please upload the response letter as a PDF file in the Supplementary materials in the online submission system. Please give detailed response. Please write down what changes have been made in the response letter, rather than reading the manuscript to find what has been revised. And please note that any uncompleted or improper corrections by the authors during this revision may lead to rejection.

1. The language needs revision by a fluent speaker, companied with a certificate of language editing service and a manuscript with tracked editing records as a Suppl.
2. Please use https://www.figcheck.com/imagecheck or similar free/paid picture duplicate checking software to detect all figures and upload the examining PDF report as a Supplement file.
3. Similar studies were found in PubMed. Authors should cite part of them and discuss what is new and different from this article.
4. The discussion is not in-depth, limitation is lack, and the clinical implications shall also be added.
5. Provide supporting references for Section Methods, especially mentioning doses, time, etc.
6. State n = ? in Section Statistical analysis and all figure legends.
7. Please provide a duplicate check report by authors as a supplementary file (total < 20%, each < 2%).


Reviewer 1 ·

Basic reporting

This paper presents a well-conducted study that explores the potential therapeutic application of Ferrostatin-1 in the treatment of acute kidney injury. The experiments are well-designed, and the results provide important insights into the role of Ferrostatin-1 in mitigating AKI. However, I have identified several areas for improvement that, if addressed, would strengthen the overall quality of the manuscript. I suggest the following amendments:

Experimental design

No comment.

Validity of the findings

No comment.

Additional comments

1. In the abstract, there appears to be a typographical error. It should read "post-regulation of Ferrostatin-1" rather than "post-regulation of androstatin-1". Please correct this to maintain the consistency of the paper.

2. In the methods section, please provide more detailed information about the dosage and administration of Ferrostatin-1 used in the mouse model. This information is crucial for evaluating the study's reproducibility and potential for translation to clinical applications.

3. The authors should include a discussion of potential side effects or toxicities associated with Ferrostatin-1 treatment. This would be important for evaluating the feasibility of this therapy in a clinical setting.

4. The in vitro experiments using HK-2 cells should be more thoroughly described, including details about cell culture conditions, treatment durations, and the methods used to assess lipid peroxidation and iron accumulation.

5. Finally, the discussion section should address the potential limitations of this study, as well as any alternative interpretations of the results. This would provide a more balanced view of the study's findings and help guide future research in this area.

Reviewer 2 ·

Basic reporting

The authors have presented an interesting and innovative study on the role of Ferrostatin-1 in attenuating acute kidney injury. The manuscript is well-written.

Experimental design

The experiments conducted are methodologically sound. However, there are some areas that require improvement and clarification to ensure the paper's scholarly standard is met.

Validity of the findings

Yes.

Additional comments

1.The title of the paper should be more precise and informative. Consider revising it to better reflect the study's findings, such as "Ferrostatin-1 Post-Treatment Attenuates Acute Kidney Injury in Mice by Inhibiting Ferritin Production and Regulating Iron Uptake-Related Proteins".

2.In the introduction, please provide a more comprehensive review of the current literature on ferritinase and its potential role in the pathophysiology of AKI. This will help contextualize your study's findings within the existing body of knowledge.

3.Provide a more detailed description of the mouse model of AKI used in the study, including its reproducibility, limitations, and relevance to the human condition.

4.The methodology section should include a thorough description of the statistical analyses employed in the study, including the specific tests used and the criteria for statistical significance.

5.In the results section, please provide the exact p-values for the findings. This will help the reader better assess the significance of the results.

---

## Round 0.2 · Minor Revisions

The article has addressed all questions from the reviewers but the Section Editor has noted the following:

> A few things need to be attended to. First, the quality of the immunoblots is very poor, and only a single experimental replicate of raw data is provided. The authors should upload all replicates for all immunoblots (and confirm these are biological and not technical replicates); this is essential.

> I also urge the authors to replace some of the blots in the figures as they are smudged and of poor quality.

> Please also carefully check you have constructed the figures with the correct blots: to my eye, GPX4 and SLC7A11 in Figure 2 look like different exposures of the same immunoblot. Please check carefully and show all replicates. The raw data should also be annotated with lane labels, a clear indication of which samples are loaded where, and the molecular weight markers noted.

> The authors then need to clarify how many biological and technical replicates were performed for each figure.

> Some of the text still needs work and more attention to detail. e.g., the authors include a reference to ethical approval in the cell culture section. This is surely in the wrong place and not relevant. There are many clumsy sentences.

Please address all the Section Editor's comments and resubmit the manuscript.

---

## Round 0.3 · accepted · Accept

It now meets the publication standard.

Reviewer 1 ·

Basic reporting

No comment.

Experimental design

No comment.

Validity of the findings

No comment.

Additional comments

No comment.

Reviewer 2 ·

Basic reporting

no comment

Experimental design

no comment

Validity of the findings

no comment

Additional comments

no comment